# Molecular and spatial epidemiology of HCV among people who inject drugs in Boston, Massachusetts

Thomas J. Stopka[1], Omar Yaghi[1], Min Li[2], Elijah Paintsil[2,3], Kenneth Chui[1], David Landy[1], Robert Heimer[2,4]*

1 Department of Public Health and Community Medicine, Tufts University School of Medicine, Boston, MA, United States of America, 2 Yale University School of Medicine, New Haven, CT, United States of America, 3 Yale University School of Public Health, New Haven, CT, United States of America, 4 Center for Interdisciplinary Research on AIDS at Yale, New Haven, CT, United States of America

* robert.heimer@yale.edu

## Abstract

Integration of genetic, social network, and spatial data has the potential to improve understanding of transmission dynamics in established HCV epidemics. Sequence data were analyzed from 63 viremic people who inject drugs recruited in the Boston area through chain referral or time-location sampling. HCV subtype 1a was most prevalent (57.1%), followed by subtype 3a (33.9%). The phylogenetic distances between sequences were no shorter comparing individuals within versus across networks, nor by location or time of first injection. Social and spatial networks, while interesting, may be too ephemeral to inform transmission dynamics when the date and location of infection are indeterminate.

## Introduction

Although now curable, hepatitis C virus (HCV) infections continue to be problematic. In the United States, HCV mortality has increased in recent years, while mortality for 61 other federally reportable infectious diseases, including HIV, has decreased [1]. HCV infections have increased by more than 70% among 15–30 year-olds during the past decade, and new infections are largely attributed to injection drug use (IDU) [1, 2]. In Massachusetts, over 2,000 new cases attributed to IDU have been reported in those <30 years of age in each of the last seven years [3]. More information about transmission networks might improve prevention efforts.

The application of phylogenetic and spatial analysis of HCV has identified a range of HCV subtypes among people who inject drugs (PWID) and provided biological evidence of disease clusters and transmission patterns in outbreaks [4–6]. We aimed to find evidence of HCV sequence alignment or transmission patterns among PWID outside of known outbreaks. As part of a study of PWID in the Boston area, we identified a subsample with active HCV infection and sequenced part of their viral genome. We then integrated behavioral, social network, spatial, and HCV phylogenetic data to explore transmission dynamics in established epidemics. We were especially interested to see if we could find evidence of HCV viral sequence alignment or transmission patterns that could identify potential targets, methods, or venues for HCV prevention and treatment interventions.

**Data Availability Statement:** All relevant data are within the manuscript and its Supporting Information files.

**Funding:** The research was funded by an internal grant from Tufts University (no number assigned) with additional funding from the Providence-Boston Center for AIDS Research (PI: Stopka), through Grant Number P30AI042853 from the National Institute of Allergy and Infectious Diseases. Support for Dr. Heimer was provided through Grant Number R01DA030420 from the National Institute on Drug Abuse. The funders had no role in study design, data collection and analysis, decision to publish, or preparation of the manuscript.

**Competing interests:** The authors have declared that no competing interests exist.

## Methods

We employed a cross-sectional design focused on PWID in the Boston area. PWID were eligible if they met the following criteria: (1) age 18–45 years; (2) English or Spanish speaking; (3) evidence of track marks; (4) living with, at risk for, or infected with and subsequently cleared HCV; and (5) ability to provide written informed consent. The study protocol was approved by the [Blinded] Institutional Review Board and the [Blinded] Human Investigation Committee. Consenting individuals completed a survey and were administered the Oraquick® HCV Rapid Antibody test (OraSure Technologies).

Between February and October 2016, recruitment relied on time-location sampling and chain referral in Boston and Cambridge. For the ***time-location sampling*** approach, trained outreach staff recruited participants through partnership with harm reduction programs, shelters, and social service agencies. Staff members at community-based agencies posted study flyers and identified PWID whom they knew were likely to have acquired HCV and would be interested in participating in the study. For the ***chain-referral approach***, enrolled participants (i.e., "seeds") were asked to refer peers within their networks to the study. In both approaches, research staff described the study to interested recruits, confirmed participant eligibility, provided written materials, and obtained written informed consent. Analysis of recruitment chains was depicted using the StatNet and UserNetR packages in R.

Participants were tested for HCV antibodies and then completed the survey on a tablet, laptop, or desktop computer in our research offices or on tablets or laptops in private spaces provided by community-based partners. Surveys covering demographic, health, and socio-behavioral topics were conducted in English or Spanish and lasted approximately 60 minutes. Participants were informed of HCV test results after completing the survey, and those who tested positive were invited to a separate appointment and were consented to have a blood sample taken. Sample collection continued until 102 individuals were enrolled in the sub-study of HCV-positive participants.

Blood specimens were centrifuged, and the serum was added to RNA*Later*® to preserve the integrity of viral RNA (RNALater; Ambion Inc., Austin, TX). Samples were stored at -80˚C. After thawing, viral RNA was purified and a 360-base region in the core gene was amplified and sequenced [7, 8]. Sequences were optimally aligned using the CLUSTAL W program and a phylogenetic tree was constructed by the Neighbor-Joining method based on Kimura's two-parameter distances (Mega software, version 7.0 [9]. We assessed the reliability of the tree topologies by bootstrapping with 100 replicates. Evolutionary distances were computed using the Maximum Composite Likelihood method [10].

We compared sequence similarity by recruitment pattern, location of injection initiation, and number of years injecting drugs. We conducted Fisher's exact tests to assess whether differences in HCV subtypes by recruitment approach were statistically significant. We subsequently ran simple logistic regression analyses to determine whether place of residence at the time of first injection or number of years injecting was significantly associated with HCV subtypes. Place of residence was divided into three categories: Boston, suburbs of Boston (i.e., outside of the city limits and on a Massachusetts Bay Transit Authority (MBTA) line), or beyond the suburbs (not reachable by MBTA line). The number of years injecting was also divided into three categories: 0–5, 6–10, >10 years.

## Results

Among the 102 participants who tested positive for HCV antibodies and provided a subsequent blood sample, 66 were actively infected and sequence data were obtained from 63 of the 66. Out of these 63, 60 completed the survey, and one additional participant completed the

survey without providing sequence data. Thus, the sample sizes are 63 for the sequence data and 61 for the survey. The mean age of the 61 respondents was 32±6 years, and males comprised 75% of the sample. Forty participants (66%) described themselves as White non-Latino, nine as Latino, eight as mixed-race or other, and four as African American. Most were single, with 42 (69%) living alone and 9 (15%) living with a partner. About three-quarters (n = 46) reported their sexual orientation as heterosexual, nine as bisexual, and six as gay or lesbian. Three-quarters of the sample reported being homeless and the median monthly income was $500–1000. Ninety percent (n = 53) reported having been incarcerated. The mean duration of IDU was 12 years (SD = 8) (Table 1).

## HCV subtype by recruitment pattern

Fig 1A shows the participant network. Thirty-one of the 63 participants were identified through chain referral, indicated by being connected with double-headed arrows, and 32 were identified through time-location sampling, indicated by nodes without connection. Genotype 1a was higher for time-location participants (56.3%; 18/32) than for chain referred participants (41.9%; 13/31; p-value: 0.211); genotype 3a was higher for chain referred participants (45.2%; 14/31) than for time-location participants (22.6%; 7/32; p-value: 0.109) [S1 Table]. While there was no exclusive sequence alignment of subtypes within the chain referral clusters, three examples of sequence alignment with n>3 only consisted of two subtypes. In addition, in the largest sequence alignment (n = 13), seven participants with subtype 3a were connected. As indicated in Fig 1A, the genetic differences are nil or very small (0.1%) within the closely aligned sequences and independent of recruitment network.

## HCV subtype by location of first injection

Among PWID with Subtype 1a, 32.4% (11/34) first injected in Boston, 20.6% (7/34) first injected in Boston suburbs, and 47.1% (16/34) first injected outside of the Boston suburbs. Among PWID with subtype non-1a, 39.1% (9/23) first injected in Boston, 30.4% (7/23) first injected in Boston suburbs, and 30.4% (7/23) first injected outside of the suburbs (S1 Table). Location of the first injection was not associated with HCV subtype, when comparing subtypes from outside of the Greater Boston Area (odds ratio [OR]: 1.87; 95% confidence interval [CI]: 0.54, 6.53) or in Boston suburbs (OR: 0.82; 95% CI: 0.21, 3.26) to subtypes from Boston.

## HCV subtype by number of years injecting

Among PWID with Subtype 1a, 21.9% (7/32) had been injecting for 0–5 years, 31.3% (10/32) had injected for 6–10 years, and 46.9% (15/32) had been injecting for >10 years. Among PWID with Subtype Non-1a, 13.7% (3/22) had been injecting for 0–5 years, 13.7% (3/22) had injected for 6–10 years, and 72.7% (16/22) had been injecting for >10 years. We divided the sample into those infected with genotype 1a versus not 1a, and there was no difference between groups by years of injection experience (OR: 0.97, 95% CI: 0.90, 1.04).

## HCV sequencing results

Sequencing results indicated that the distances between sequences in phylogenetic analyses were no shorter comparing sequence data within versus across networks. Four of the oldest injectors in the sample (individuals born in 1977 or earlier) were infected with HCV genotypes 1b and 2b. Only one other individual was infected with either of these two subtypes (Fig 1B). But among closely related sequences with genotypes 1a and 3a, network connections did not predict sequence homology.

**Table 1. Descriptive statistics: People who inject drugs in the Greater Boston Area of Massachusetts, 2016 (n = 61).**

|  | Total |
|---|---|
|  | N = 61 |
| Age (year); mean (standard deviation) | 32 (6) |
| Gender |  |
| Male | 46 (75%) |
| Female | 15 (25%) |
| Race/ethnicity |  |
| African American | 4 (7%) |
| American Indian/Alaskan Native | 1 (2%) |
| Hispanic, Latino, or Latina | 9 (15%) |
| White, non-Hispanic | 40 (66%) |
| Mixed races & others | 7 (11%) |
| Marital status |  |
| Single/living alone | 42 (69%) |
| Single, living w/ partner | 9 (15%) |
| Married/ in domestic partnership | 2 (3%) |
| Divorced | 3 (5%) |
| Separated | 3 (5%) |
| Widowed | 2 (3%) |
| Sexual orientation |  |
| Heterosexual/straight | 46 (75%) |
| Gay/lesbian | 6 (10%) |
| Bisexual | 9 (15%) |
| Education level |  |
| Grade 6–8 | 3 (5%) |
| Grade 9–12, not HS grad | 14 (23%) |
| HS grad/GED | 32 (52%) |
| Some colleage, no degree | 9 (15%) |
| 2–4 yrs college degree | 1 (2%) |
| Graduate/professional degree | 2 (3%) |
| Homeless |  |
| No | 15 (25%) |
| Yes | 46 (75%) |
| Main residence, last 30 days |  |
| Own apartment, house, or room | 7 (11%) |
| Home of parents, relatives, or friends | 10 (16%) |
| Halfway house, group home, or foster home | 6 (10%) |
| Hotel or motel | 2 (3%) |
| Shelter | 18 (30%) |
| Abandoned building | 2 (3%) |
| Public park | 4 (7%) |
| Street, wooded area | 7 (11%) |
| Other | 5 (8%) |
| Income, last 30 days |  |
| None | 7 (11%) |
| < $500 | 17 (28%) |
| $500 to < $1000 | 22 (36%) |

*(Continued)*

**Table 1.**  (Continued)

|  | Total |
| --- | --- |
| $1000 to < $2000 | 8 (13%) |
| $2000 to < $4000 | 4 (7%) |
| $4000 to < $6000 | 1 (2%) |
| > = $6000 | 2 (3%) |
| Ever in jail, prison, JDC |  |
| No | 6 (10%) |
| Yes | 55 (90%) |
| Currenlty on parole |  |
| No | 53 (87%) |
| Yes | 8 (13%) |

## Discussion

Integrating social network, spatial, and molecular data from established epidemics poses unique challenges. These stem from the lack of defined dates of infection, the durability of current social network connections, the multilevel nature of risk environments, and missing data about the full network structure that is obtained when chain referral relationships are the sole data source on social connections [11–13]. For this reason, we sought to determine if HCV was more closely related (1) when specimens were obtained from within recruitment chains than those obtained by the less directed recruitment achieved by time-location sampling, and (2) when participants were categorized by time and location that their first exposure to HCV had likely have occurred. For the latter analysis, we made the working hypothesis that HCV infection is an early consequence of initiating injection drug use. This hypothesis is consistent with observations of high HCV incidence in the first year of injection.

Our findings that HCV subtype 1a was the predominant subtype among PWID in the Boston Area (57.1%), followed by subtype 3a (33.9%), differs from studies elsewhere that focused on HCV among PWID. A study from New York, conducted twenty years earlier than ours, found that the predominant genotypes were 1a and 1b, with 3a comprising only 5% of sequences genotyped [14]. A study from Baltimore, concurrent with ours, found no evidence of genotype 3a [15]. Our study population was substantially younger than those in the other two studies, consistent with our aim to recruit a younger sample, but differences in genotype were not associated with age. However, there were two subtypes (1b and 2b) that were restricted to older individuals. It appears, however, that these subtypes have failed to spread among younger members of our study population. Analysis of HCV genotypes from PWID in Vancouver revealed a pattern similar to ours, with nearly half their sample infected with genotype 1a and one-third infected with genotype 3a, but the relevance of this finding is limited by the geographic distance between coasts [6].

HCV phylogenetic sequence alignment has been found in those recently initiating drug injection. Sequence alignment was common in the large HIV-HCV outbreak in Scott County, Indiana. But even there, multiple introductions of HCV were identified. Genotypes 1a and 3b accounted for 72% and 20% of the patient samples, respectively, but additional genotypes had entered the mix, and some people were infected with multiple genotypes [16]. One study from Melbourne found within-network correlation with HCV genotype, but only in individuals infected with genotype 3 and reporting recent drug injection initiation [17]. In our study, we did not find any significant associations between HCV subtypes by recruitment types, network characteristics, number of years injecting, nor location of first injection.

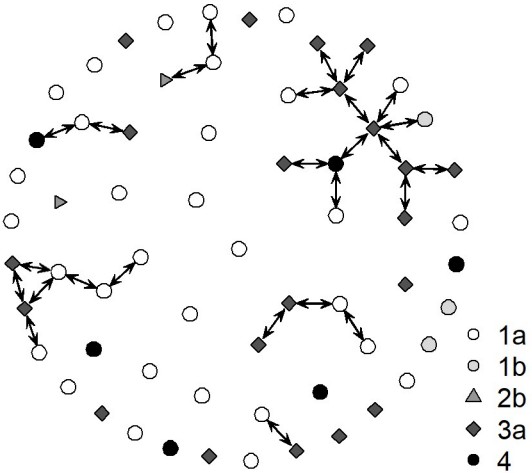

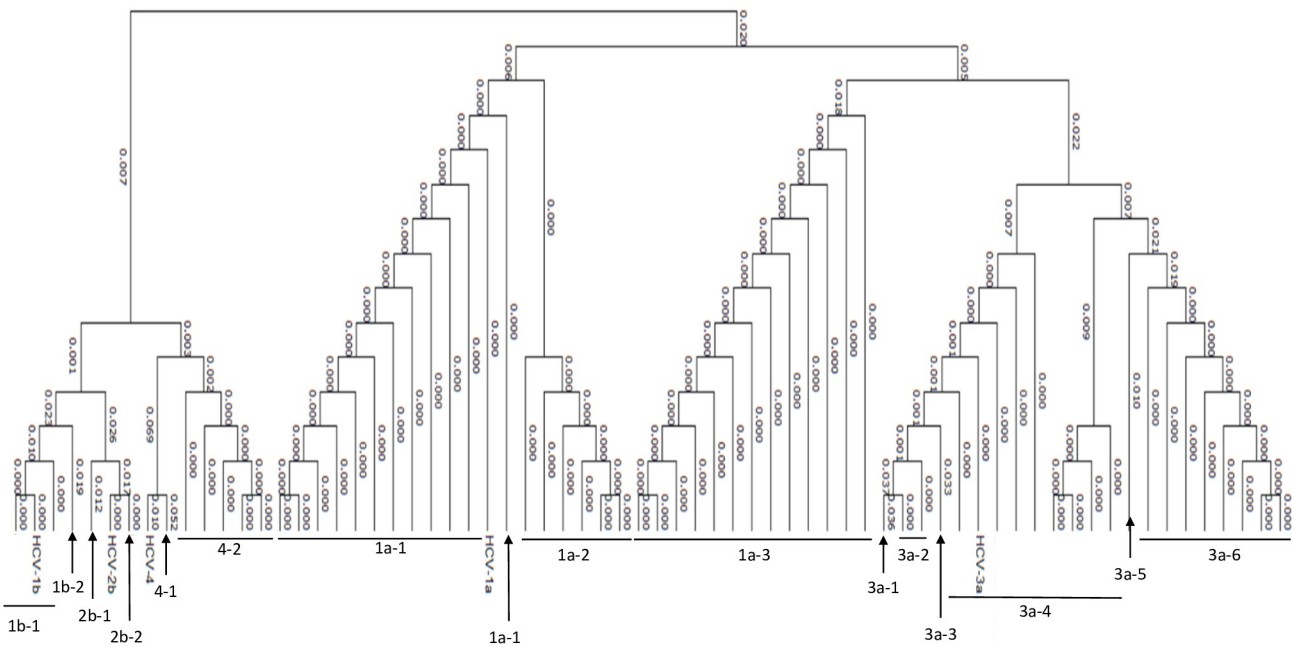

**Fig 1.** A) Network diagram showing the referral network. The different node symbols represent the five major hepatitis C virus (HCV) subtypes. The double-headed arrows indicate connection by referral (n = 31). This provides visual evidence that the social network that was accessed to recruit participants contained people infected with different strains of HCV. B) Phylogenetic tree for HCV positive people who inject drugs in Boston, Massachusetts, 2016 (n = 63). Genetic distance, reported as the proportion of sequence divergence between connections on the tree. Each subgenotype aligns closely, but the relatively short core region of the genome that we chose to sequence and the small number of specimens from people infected with genotype 1b resulted in genotype 1b sequences appearing more closely aligned with genotype 2 and 4 than with genotype 1a sequences.

Several limitations should be considered in interpreting our findings. First, our sample is small and needs to be expanded. Second, while we aimed to use respondent driven sampling (RDS), we ultimately used a modified RDS approach combined with a time-location sampling approach to recruit our sample during a relatively tight timeframe based on funding requirements. Third, we used self-reported measures from our survey, and questions referring to

events from the distant past (place where first injected, date of first injection), could have introduced recall bias. Fourth, our results are not generalizable to the entire Boston Area, nor to other cities. Finally, although we sought to recruit a relatively young sample of PWID, many people had long injecting experiences, so their social networks at the time they were infected with HCV may have been different from their social networks at the time of the survey. In lieu of trying to design studies that collect detailed individual network and phylogenetics to reconstruct patterns of disease transmission, it may be more reasonable to collect information sufficient to populate random mixing and other forms of stochastic models [18]. Sequencing of infectious disease transmission among PWID may be most effective in the case of disease outbreaks [19].

## Conclusion

Social networks, while interesting, are too ephemeral to inform transmission dynamics if the date and location of infection are indeterminate. Expanding research efforts to obtain extensive social network data from PWID populations in established epidemics may not be worthwhile or cost-effective given the expense of obtaining full social network data. Alternatives, such as quasi network data obtained through RDS and random mixing models, may be sufficient when modeling transmission networks.

## Supporting information

**S1 Table. HCV subtype by town of first injection and number of years injecting, Boston and Cambridge, Massachusetts, 2016.**
(DOCX)

**S1 File.**
(PDF)

## Acknowledgments

The authors wish to acknowledge the assistance of the following agencies and individuals who contributed to recruitment and data collection efforts through participant referrals or provision of space for data collection activities: AHOPE, AIDS Action Committee, South Boston Behavioral Health Center, Dr. Laura Grubb, Tufts Medical Center, and St. Anthony's Shelter. The authors also wish to thank Dr. Nadia Abdala for guidance on HCV specimen sequencing and preparation for phylogenetic analyses.

## Author Contributions

**Conceptualization:** Thomas J. Stopka, Kenneth Chui, Robert Heimer.

**Data curation:** Thomas J. Stopka, Min Li, David Landy.

**Formal analysis:** Thomas J. Stopka, Elijah Paintsil, David Landy, Robert Heimer.

**Funding acquisition:** Thomas J. Stopka.

**Investigation:** Thomas J. Stopka, Omar Yaghi.

**Methodology:** Thomas J. Stopka, Omar Yaghi, Min Li, Elijah Paintsil, Kenneth Chui.

**Supervision:** Thomas J. Stopka, David Landy.

**Writing – original draft:** Thomas J. Stopka, Elijah Paintsil, Kenneth Chui, Robert Heimer.

**Writing – review & editing:** Thomas J. Stopka, Omar Yaghi, Elijah Paintsil, Kenneth Chui, David Landy, Robert Heimer.

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
