## [Decision Letter · Decision Letter 0]

22 Apr 2022

PONE-D-22-07761Molecular and spatial epidemiology of HCV among people who inject drugs in Boston, MassachusettsPLOS ONE

Dear Dr. Heimer,

Thank you for submitting your manuscript to PLOS ONE. After careful consideration, we feel that it has merit but does not fully meet PLOS ONE’s publication criteria as it currently stands. Therefore, we invite you to submit a revised version of the manuscript that addresses the points raised during the review process.

The two reviewers made many constructive comments by which your manuscript could be improved. Please address all of them before resubmitting, in particular by also making sequence data available on public repositories. ==============================

We look forward to receiving your revised manuscript.

Kind regards,

Joël Mossong

Academic Editor

PLOS ONE

Journal Requirements:

"This research was funded by a 2016 grant from the Tufts Institute for Innovation (PI: Stopka). Additional support was provided by the Providence-Boston Center for AIDS Research (PI: Stopka), through Grant Number P30AI042853 from the National Institute of Allergy and Infectious Diseases. Support for Dr. Heimer was provided through Grant Number R01DA030420 from the National Institute on Drug Abuse. The content is solely the responsibility of the authors and does not necessarily represent the official views of the National Institute of Allergy and Infectious Diseases or the National Institute of Health."

We note that you have provided funding information. However, funding information should not appear in the Funding section or other areas of your manuscript. We will only publish funding information present in the Funding Statement section of the online submission form. 

"The research was funded by an internal grant from Tufts University (no number assigned) with additional funding from the Providence-Boston Center for AIDS Research (PI: Stopka), through Grant Number P30AI042853 from the National Institute of Allergy and Infectious Diseases. Support for Dr. Heimer was provided through Grant Number R01DA030420 from the National Institute on Drug Abuse. 

Initials of the authors who received each award

TJS for Tufts Institute for Innovation (TII), which is now defunct

The TII award did not have a grant number; internal mechanism

The full name of each funder

See above and below

URL of each funder website

TII no longer exists; no website

Additional support from the Providence-Boston CFAR: https://cfar.med.brown.edu/ P30AI042853 from NIAiD; https://www.niaid.nih.gov/

RH received support through R01DA030420 from NIDA. NIDA: https://nida.nih.gov/

Did the sponsors or funders play any role in the study design, data collection and analysis, decision to publish, or preparation of the manuscript? No

Reviewers' comments:

Reviewer's Responses to Questions

**Comments to the Author**

1. Is the manuscript technically sound, and do the data support the conclusions?

Reviewer #1: Partly

Reviewer #2: Yes

2. Has the statistical analysis been performed appropriately and rigorously? 

Reviewer #1: Yes

Reviewer #2: Yes

3. Have the authors made all data underlying the findings in their manuscript fully available?

Reviewer #1: Yes

Reviewer #2: Yes

4. Is the manuscript presented in an intelligible fashion and written in standard English?

Reviewer #1: Yes

Reviewer #2: Yes

5. Review Comments to the Author

Reviewer #1: In their manuscript, “Molecular and special epidemiology of HCV among who inject drugs in Boston, Massachusetts”, the authors use survey data from 61 individuals and sequence data from 63 individuals in attempt to find evidence of viral clustering and factors associated with that clustering for potential targets for intervention. Unfortunately, their survey size was entirely too small for the goals of the research presented, namely “We aimed to find evidence of disease clusters or transmission patterns outside of known outbreaks” lines 66-67. Otherwise this is a very good study and the paper is well written. The limitations section was particularly well written and enlightening.

There are a couple issues to address to clarify the study.

1) The conclusion in the abstract, “Social and special networks, while interesting, may be too ephemeral to inform transmission dynamics when the date and location of infection are indeterminant”, is not supported by the data presented. The data presented simply does not have the depth of sampling to explore transmission dynamics of HCV in Boston. The authors estimate 14,000 new HCV infections over the last seven years (line 60) and the average duration of infection of the 102 individuals recruited was 12 years. The authors, were able to survey about <1% of HCV infected individuals in their population. It would seem incredibly unlikely that any clusters or evidence of transmission patterns would be identified in such a small survey size.

2) The authors spent 9 months (February to October 2016, line 82) recruiting subjects. Why was there such a low yield in study subjects?

3) How many individuals were surveyed to identify the 102 HCV antibody positive subjects? How many individuals were approached? How many individuals had their blood tested?

4) The authors state in their method that a phylogenetic tree was generated (lines 105 to 110) and provide a figure of the tree (Figure 1B), however, there is no presentation of the results It is unclear if there was any viral clustering. Were there any viral clusters? The figure legend needs more information. It is not clear what the values of on the phylogenetic tree mean.

Reviewer #2: In this study, the authors aim to see if phylogenetic analysis, when combined with behavioural, network and spatial data, can add to the understanding of transmission patterns among PWID outside of outbreak settings. I think the topic is interesting, and the manuscript is well written but having said that, I do think the sample size used for the study was small. I wonder if re-framing the study slightly, to be clearer that the authors were trying to see if this recruitment and analysis approach can be informative at uncovering disease clusters and transmission patterns might help a bit more – currently this is only mentioned at the end of the article in the conclusions.

Minor comments

• Line 103/4 – it would be helpful if the authors referenced where they got the sequencing protocol they used here. More detail on genotyping methods would be useful – I note the authors mention mixed genotype infections in the discussion but didn’t say if they found any themselves. Did the genotyping method mean they were unable to identify these?

• Is there a particular reason the core region was chosen for this analysis – is it the best region for this type of analysis?

• HCV sequencing results and comparison – is gt1a vs non gt1a a reasonable comparison? From my understanding, the spread of gt3a among PWID has been more recent than gt1a and seems consistent with the observation that older individuals in the cohort were infected with gt1b/2b. Whilst numbers are low, I wonder if the authors would see more trends associated with age/location/IDU duration if they further split the non-1a group?

• Perhaps consider adding a table with some of the data from lines 133-164 comparing people with different HCV genotypes (1a, 3a, other genotypes?) – might make all the information easier to digest for readers

• Line 200 - given the authors were aiming to look at transmission clusters between PWID outside of known outbreaks, I agree the sample size is small particularly outside of an outbreak context. The authors say it needs to be expanded – I would be interested to know what they think a reasonable size would be?

• I think some discussion around the issue that phylogenetic linkage will only be possible for samples with the same genotype is missing. Did the authors consider if there was a better way to sample the population to get a larger cohort?

• Table 1, is the age mean or median?

• Could the authors please confirm they will upload the HCV sequences into a sequence repository and cite these in the manuscript if accepted.

Typos

Line 64/5; analysis of HCV has identified [a range of] HCV subtypes among

Line 126; Forty [participants] (66%) described themselves as

6. PLOS authors have the option to publish the peer review history of their article (what does this mean?). If published, this will include your full peer review and any attached files.

Reviewer #1: No

Reviewer #2: No

---

## [Author Response · Author response to Decision Letter 0]

20 Jun 2022

Reviewer 1 Comments:

Reviewer #1: In their manuscript, “Molecular and special epidemiology of HCV among who inject drugs in Boston, Massachusetts”, the authors use survey data from 61 individuals and sequence data from 63 individuals in attempt to find evidence of viral clustering and factors associated with that clustering for potential targets for intervention. Unfortunately, their survey size was entirely too small for the goals of the research presented, namely “We aimed to find evidence of disease clusters or transmission patterns outside of known outbreaks” lines 66-67. Otherwise this is a very good study and the paper is well written. The limitations section was particularly well written and enlightening.

Response: Thank you for the positive feedback on the writing and enlightening limitations sections of our manuscript. While a typical survey with a sample size such as ours (n=61) would be considered small for typical analyses, phylogenetic analyses introduce an additional level of complexity, and there are few studies in the US that have large samples sizes that facilitate assessment of phylogenetic clusters. Admittedly, future studies, with larger funding levels should aim to recruit, enroll, and analyze larger samples, but our study is one of only a few in the Northeastern US that have begun to employ combined spatial and phylogenetic analyses. Please also see our related response to the first comment from Reviewer #2 below. 

There are a couple issues to address to clarify the study.

1) The conclusion in the abstract, “Social and special networks, while interesting, may be too ephemeral to inform transmission dynamics when the date and location of infection are indeterminant”, is not supported by the data presented. The data presented simply does not have the depth of sampling to explore transmission dynamics of HCV in Boston. The authors estimate 14,000 new HCV infections over the last seven years (line 60) and the average duration of infection of the 102 individuals recruited was 12 years. The authors, were able to survey about <1% of HCV infected individuals in their population. It would seem incredibly unlikely that any clusters or evidence of transmission patterns would be identified in such a small survey size.

Response: Our study was relatively small, by design, as an exploratory mixed methods study, and we met our recruitment goals. The comments from Reviewer 1 regarding the external validity of our study may have some merit. It is also important to bear in mind that we don’t have a definition for a cluster. We were looking for relative sequence similarity, acknowledging that people in a linked social network would have more similarity in their sequences than people in a chain who were not connected through a social network. We have removed the term “cluster” from our manuscript and replaced it with “sequence alignment” throughout the manuscript. We have also defined the genetic distance terminology. There is no concordance between the network structure of recruitment and the genetic distance. 

2) The authors spent 9 months (February to October 2016, line 82) recruiting subjects. Why was there such a low yield in study subjects?

Response: Our overall study incorporated a mixed methods approach, including qualitative in-depth interviews, development and fielding of an Audio Computer Assisted Self Interview (ACASI), piloting of a text messaging (chatbot) system, HCV rapid testing, phlebotomy, specimen preparation, PCR testing, phylogenetic and bioinformatics analyses and additional study components. Thus, as part of a mixed methods exploratory study, we completed all of our target recruitment goals for our qualitative (n=24) [https://pubmed.ncbi.nlm.nih.gov/30071457/], ACASI survey (n=252) [https://pubmed.ncbi.nlm.nih.gov/29482054/], HCV testing (n=102), and phylogenetic analyses (n=62). This was the first such mixed methods study conducted with PWID in the Boston Area, and as an exploratory study, we were interested in assessing the relationship between social network structure and phylogenetic similarity, and we achieved this goal.

3) How many individuals were surveyed to identify the 102 HCV antibody positive subjects? How many individuals were approached? How many individuals had their blood tested?

Response: We aimed to conduct HCV rapid testing until we achieved identification of 100 study participants who tested HCV antibody positive. While we surveyed 252 individuals using our ACASI instrument, we limited the number of participants who would undergo confirmatory blood draws and phylogenetic analyses given the costs and the limited budget of our pilot project. We ultimately obtained blood specimens from 102 participants. Among these 102 participants, 40% had cleared the virus or did not have sufficient virus to be sequenced. We could only confirm chronic infection in 63 participants and 61 yielded readable sequences. We don’t know how many individuals were approached as the protocol did not require that we keep information on potential recruits who were not ultimately consented.

4) The authors state in their method that a phylogenetic tree was generated (lines 105 to 110) and provide a figure of the tree (Figure 1B), however, there is no presentation of the results. It is unclear if there was any viral clustering. Were there any viral clusters? The figure legend needs more information. It is not clear what the values of on the phylogenetic tree mean.

Response: Given word count limitations, we were succinct in our results section, but we acknowledge that additional details are merited. We have now added additional text in our results section and in the legend for Figure 1B to denote that genetic distances are reported. 

Reviewer #2: In this study, the authors aim to see if phylogenetic analysis, when combined with behavioural, network and spatial data, can add to the understanding of transmission patterns among PWID outside of outbreak settings. I think the topic is interesting, and the manuscript is well written but having said that, I do think the sample size used for the study was small. I wonder if re-framing the study slightly, to be clearer that the authors were trying to see if this recruitment and analysis approach can be informative at uncovering disease clusters and transmission patterns might help a bit more – currently this is only mentioned at the end of the article in the conclusions.

Response: Thank you for the positive feedback. Considering the first comments from both Reviewers 1 and 2, we have reframed the manuscript slightly, per recommendations, clarifying that our study aimed to determine whether recruitment and analysis approaches can be informative at uncovering disease sequences and transmission patterns, better aligning our intro, methods, results and discussion. We were looking to determine if there was an alignment of sequences among individuals who appeared to be linked to each other/know each other based on recruitment patterns compared to others for whom we did not have linkage data. Although we were mildly disappointed to discover that this was not the case, our discussion helps to contextualize our findings in the larger literature and how our approach can help to inform future studies. 

Minor comments

• Line 103/4 – it would be helpful if the authors referenced where they got the sequencing protocol they used here. More detail on genotyping methods would be useful. 

Response: We have added references for sequencing of HCV core region and sequence alignment as suggested by the reviewer.

– I note the authors mention mixed genotype infections in the discussion but didn’t say if they found any themselves. Did the genotyping method mean they were unable to identify these?

Response: We don’t think so, but we only performed consensus and not deep sequencing. 

• Is there a particular reason the core region was chosen for this analysis – is it the best region for this type of analysis?

Response: The core region is a very conservative region and quite popular in these types of analyses. Like others, we used this approach in a previous study in St. Petersburg, Russia (citation #8).

• HCV sequencing results and comparison – is gt1a vs non gt1a a reasonable comparison? From my understanding, the spread of gt3a among PWID has been more recent than gt1a and seems consistent with the observation that older individuals in the cohort were infected with gt1b/2b. Whilst numbers are low, I wonder if the authors would see more trends associated with age/location/IDU duration if they further split the non-1a group?

Response: We conducted these analyses employing a number of approaches. We re-ran the analyses of the 3a genotypes, but we found no differences. We have edited a sentence to the manuscript to clarify that we looked but did not find any network characteristics associated with genotypes. (p. 7, lines: 216-217). 

• Perhaps consider adding a table with some of the data from lines 133-164 comparing people with different HCV genotypes (1a, 3a, other genotypes?) – might make all the information easier to digest for readers

Response: Since brief reports only allow 2 tables and figures, we refrained from creating additional tables and figures. However, we now include a supplemental table, per the reviewer’s recommendation, while retaining the existing table and figure. 

• Line 200 - given the authors were aiming to look at transmission clusters between PWID outside of known outbreaks, I agree the sample size is small particularly outside of an outbreak context. The authors say it needs to be expanded – I would be interested to know what they think a reasonable size would be?

Response: We have removed mentions of clusters throughout the manuscript. We do not have an explicit definition of the closeness of a sequence to be considered a cluster, so we can’t really answer this question. As the reviewer noted, we obtained sequences from a very small subset of people in Boston infected with HCV, so we have no information on how the 61 sequences we have align with we vast array of virus in circulation among PWID in the Boston area. Thus, we do not have the necessary data to make such an estimate.

• I think some discussion around the issue that phylogenetic linkage will only be possible for samples with the same genotype is missing. Did the authors consider if there was a better way to sample the population to get a larger cohort?

Response: The purpose of this exploratory study was to determine what resources would be necessary to obtain a fuller picture of HCV transmission dynamics. Unless we have a way to link rapid deep sequencing to recruitment patterns, the effort is unlikely to identify recent outbreaks in a timely fashion and to an extent that is superior to simply enhancing widespread HCV testing in the focal/key populations who inject drugs. This would require collaboration with other sites and pooling data. 

• Table 1, is the age mean or median?

Response: Mean age and standard deviation. We have updated wording in the table to make this clear.

• Could the authors please confirm they will upload the HCV sequences into a sequence repository and cite these in the manuscript if accepted.

Response: Yes, we have submitted the sequences to GenBank and they are under review. We will provide the accession number can confirm that Dr. Paintsil has the sequences and will upload them if the manuscript is accepted for publication. 

Typos

Line 64/5; analysis of HCV has identified [a range of] HCV subtypes among

Line 126; Forty [participants] (66%) described themselves as

Response: Thank you for catching these. We have corrected the typos.

---

## [Decision Letter · Decision Letter 1]

4 Jul 2022

PONE-D-22-07761R1Molecular and spatial epidemiology of HCV among people who inject drugs in Boston, MassachusettsPLOS ONE

Dear Dr. Heimer,

Thank you for submitting your manuscript to PLOS ONE. After careful consideration, we feel that it has merit but does not fully meet PLOS ONE’s publication criteria as it currently stands. Therefore, we invite you to submit a revised version of the manuscript that addresses the points raised during the review process.

ACADEMIC EDITOR:One reviewer still has some minor comments. Please address all of these. ==============================

We look forward to receiving your revised manuscript.

Kind regards,

Joël Mossong, PhD

Academic Editor

PLOS ONE

Journal Requirements:

Reviewers' comments:

Reviewer's Responses to Questions

**Comments to the Author**

1. If the authors have adequately addressed your comments raised in a previous round of review and you feel that this manuscript is now acceptable for publication, you may indicate that here to bypass the “Comments to the Author” section, enter your conflict of interest statement in the “Confidential to Editor” section, and submit your "Accept" recommendation.

Reviewer #2: All comments have been addressed

2. Is the manuscript technically sound, and do the data support the conclusions?

Reviewer #2: Yes

3. Has the statistical analysis been performed appropriately and rigorously? 

Reviewer #2: Yes

4. Have the authors made all data underlying the findings in their manuscript fully available?

Reviewer #2: Yes

5. Is the manuscript presented in an intelligible fashion and written in standard English?

Reviewer #2: Yes

6. Review Comments to the Author

Reviewer #2: I am happy with the authors responses to my comments. After reviewing the updates I have a couple more minor queries relating to Figure 1B –

I note that genotype 1b sequences cluster with gt2/4 sequences, rather than with gt1a sequences – I assume this is because core is being used rather than NS5B or whole genome? It might be worth commenting on this in the figure legend for those less familiar with HCV phylogenies if so.

Could the authors add more information on what exactly the labels on the tree mean into the figure legend? I initially assumed they were labelling HCV subtypes, but I don’t think this is correct. If they are highlighting more closely related sequences, could they please list the criteria (ie distance threshold) for doing this?

7. PLOS authors have the option to publish the peer review history of their article (what does this mean?). If published, this will include your full peer review and any attached files.

Reviewer #2: No

---

## [Author Response · Author response to Decision Letter 1]

27 Jul 2022

Responses to Reviewer #2

Reviewer #2: I am happy with the authors responses to my comments. After reviewing the updates I have a couple more minor queries relating to Figure 1B –

I note that genotype 1b sequences cluster with gt2/4 sequences, rather than with gt1a sequences – I assume this is because core is being used rather than NS5B or whole genome? It might be worth commenting on this in the figure legend for those less familiar with HCV phylogenies if so. 

We have added this point to the figure legend, which now reads: “Genetic distance, reported as the proportion of sequence divergence between connections on the tree. Each sub-genotypes aligns closely, but the relatively short core region of the genome that we chose to sequence and the small number of specimens from people infected with genotype 1b resulted in genotype 1b sequences appearing more closely aligned with genotype 2 and 4 than with genotype 1a sequences.”

Could the authors add more information on what exactly the labels on the tree mean into the figure legend? I initially assumed they were labelling HCV subtypes, but I don’t think this is correct. If they are highlighting more closely related sequences, could they please list the criteria (ie distance threshold) for doing this?

We address this point in the revision by amending the text to read: “The double-headed arrows indicate connection by referral (n=31). This provides visual evidence that the social network that was accessed to recruit participants contained people infected with different strains of HCV.”

---

## [Editor Report · Decision Letter 2]

1 Aug 2022

Molecular and spatial epidemiology of HCV among people who inject drugs in Boston, Massachusetts

PONE-D-22-07761R2

Dear Dr. Heimer,

We’re pleased to inform you that your manuscript has been judged scientifically suitable for publication and will be formally accepted for publication once it meets all outstanding technical requirements.

Kind regards,

Joël Mossong, PhD

Academic Editor

PLOS ONE
---

## [Editor Report · Acceptance letter]

9 Aug 2022

PONE-D-22-07761R2 

Molecular and spatial epidemiology of HCV among people who inject drugs in Boston, Massachusetts 

Dear Dr. Heimer:

I'm pleased to inform you that your manuscript has been deemed suitable for publication in PLOS ONE. Congratulations! Your manuscript is now with our production department. 

Kind regards, 

on behalf of

Dr. Joël Mossong 

Academic Editor

PLOS ONE